# Catecholamines reduce choice history biases in perceptual decision making

Jan Willem de Gee[1,2,3]*, Niels A. Kloosterman[1,4,5,6], Anke Braun[1,7], Tobias H. Donner[1]*

1 Department of Neurophysiology and Pathophysiology, University Medical Center Hamburg-Eppendorf, Hamburg, Germany, 2 Cognitive and Systems Neuroscience, Swammerdam Institute for Life Sciences, University of Amsterdam, Amsterdam, The Netherlands, 3 Amsterdam Brain & Cognition, University of Amsterdam, Amsterdam, The Netherlands, 4 Department of Psychology, University of Lübeck, Lübeck, Germany, 5 Center of Brain, Behavior and Metabolism, University of Lübeck, Lübeck, Germany, 6 Max Planck Institute for Human Development, Berlin, Germany, 7 Department of Psychiatry and Neurosciences, Charité—Universitätsmedizin Berlin, Berlin, Germany

* j.w.degee@uva.nl (JWdG); t.donner@uke.edu (THD)

## Abstract

Theoretical accounts postulate that the catecholaminergic neuromodulator noradrenaline shapes cognition and behavior by reducing the impact of prior expectations on learning, inference, and decision-making. A ubiquitous effect of dynamic priors on perceptual decisions under uncertainty is choice history bias: the tendency to systematically repeat, or alternate, previous choices, even when stimulus categories are presented in a random sequence. Here, we directly test for a causal impact of catecholamines on these priors. We pharmacologically elevated catecholamine levels in human participants through the application of the noradrenaline reuptake inhibitor atomoxetine. We quantified the resulting changes in observers' history biases in a visual perceptual decision task. Choice history biases in this task were highly idiosyncratic, tending toward choice repetition or alternation in different individuals. Atomoxetine decreased these biases (toward either repetition or alternation) compared to placebo. Behavioral modeling indicates that this bias reduction was due to a reduced bias in the accumulation of sensory evidence, rather than of the starting point of the accumulation process. Atomoxetine had no significant effect on other behavioral measures tested, including response time and choice accuracy. Atomoxetine and variations of pupil-linked arousal at slower and faster timescales had analogous effects on choice history bias. We conclude that catecholamines reduce the impact of a specific form of prior on perceptual decisions.

## Introduction

Biases are prevalent in human decisions ranging from those based on abstract reasoning to judgments of the sensory environment [1–3]. One prominent source of such biases in perceptual decisions across mammalian species is the history

**Data availability statement:** All resources, including data and code used for the analyses in this paper (in Python), are publicly available at https://doi.org/10.5281/zenodo.16779024.

**Funding:** This work was supported by Amsterdam Brain and Cognition (Project grant to THD), the German Research Foundation (DO 1240/3-1 and SFB 936 to THD), and the Dutch Research Council (VI.Veni.232.210 to JWG) The funders had no role in study design, data collection and analysis, decision to publish, or preparation of the manuscript.

**Competing interests:** The authors have declared that no competing interests exist.

**Abbreviations :** ECG, electrocardiogram; MEG, magnetoencephalography; RT, reaction time.

of previous experiences [4–14]. When, as is common in natural environments, the environmental state evaluated is relatively stable, such history biases reflect adaptive expectations that are dynamically updated over time and improve performance [5,15,16]. For example, one might choose to forage for deer in the same area where previous expeditions were successful. While recent work in rodents and primates has identified correlates of such history biases in neural population activity of parietal and frontal cortical areas [9,12,17–21], little is currently known about the neurotransmitter systems involved in this important form of bias.

Influential theoretical work postulates that the catecholaminergic neuromodulator noradrenaline shapes behavior by reducing the impact of prior expectations on learning, inference, and decision-making [22,23]. The catecholaminergic (and in particular, the noradrenaline) systems of the brainstem project to large parts of the cerebral cortex, where the neuromodulators they release change the functional properties of their target networks [24–29]. Consequently, they are in an ideal position to shape cortex-wide network activity underlying decision-making in a coordinated fashion. Indeed, pupil responses (a peripheral proxy of central arousal state [30]) point to a role of neuromodulation in regulating the impact of these history-dependent priors [8]. However, pupil responses reflect the activity of multiple neuromodulatory brainstem systems [30–32], and the above result builds on correlative approaches. Direct evidence for the role of catecholaminergic neuromodulation in choice history biases is currently sparse in animal models [13,33] and absent in humans.

Here, we aimed to provide causal evidence for the role of catecholamines in regulating (suppressing) history biases in human choice behavior. Specifically, we hypothesized that catecholamines downregulate the impact of previous experiences on decision-making, which should translate into a reduction of choice history bias. We tested this hypothesis by pharmacologically increasing central catecholamine (in particular noradrenaline) levels through the selective noradrenaline reuptake inhibitor atomoxetine [34,35] and quantified the resulting behavioral changes in a variant of a visual perceptual decision task adopted from recent animal physiology [36]. Atomoxetine inhibits the presynaptic norepinephrine transporter, preventing the reuptake of norepinephrine throughout the brain along with inhibiting the reuptake of dopamine in specific brain regions such as the prefrontal cortex [34,35]. The administration of atomoxetine at the dosage used in the current study (40 mg) produces robust central effects on several neurophysiological markers of the state of cortical networks: suppression of low-frequency spectral power [37] as well as changes in long-range temporal correlations of local amplitude envelopes [37], in inter-area correlations during task [38] and in signatures of evidence accumulation [39,40]. We found that atomoxetine reduced choice history biases by specifically interacting with the way people accumulate noisy perceptual evidence.

## Results

Atomoxetine was administered orally (40 mg per session) in a within-subject, double-blind, placebo-controlled, and randomized design (Fig 1A). The behavioral task entailed the presentation of visual targets (Gabor patch of fixed orientation and

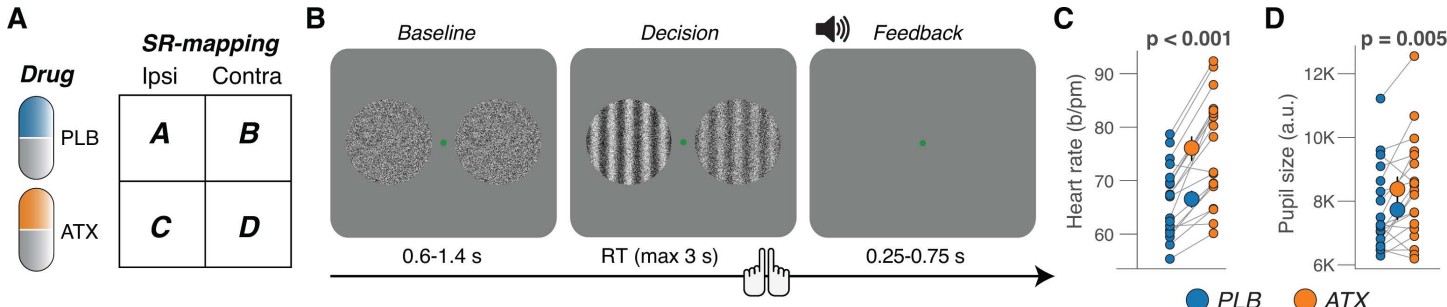

**Fig 1. Experimental design and effects on peripheral arousal markers. (A)** Design was within-subject, double-blinded, randomized, and placebo-controlled (Materials and methods). Additionally, the stimulus-response mapping was counterbalanced within participants (S1A Fig; Materials and methods). **(B)** Schematic sequence of events during the spatial contrast discrimination task (Materials and methods). Participants reported the location of the highest contrast Gabor patch (difference in contrast is high for illustration only). **(C)** Heart rate measured as the average across task blocks (Materials and methods), separately for placebo and atomoxetine sessions. Every connecting line is a participant; large data points in the middle are the group averages (error bars, s.e.m. across 19 participants); stats, paired-sample $t$ test. **(D)** As C, but for pupil size measured as the average across task blocks (Materials and methods). https://doi.org/10.5281/zenodo.16779024.

varying contrast), one in the left and one in the right visual hemifield; participants were asked to report the location of the larger contrast target by button press (Fig 1B; Materials and methods). To increase sensory uncertainty and promote integration of evidence across time [41], we added dynamic visual noise to both Gabor patches (Fig 1B; Materials and methods). There ser two different signal strengths (small and large differences in contrast); auditory feedback was presented 50 ms right after participant's choices. We collected many trials per participant (range, 3,520–4,320) across four experimental sessions, in which we systematically varied and counterbalanced drug condition and stimulus-response mapping (Figs 1A and S1A; Materials and methods).

Atomoxetine administration at this relatively low dose had a measurable effect on the peripheral arousal markers heart rate (Fig 1C) and pupil size (Fig 1D). Pupil size is a measure of central arousal [30]. Nonluminance-mediated changes of pupil diameter are causally related to the activity of neuromodulatory centers in the brainstem, including the noradrenergic locus coeruleus [31,42–44]. There was no statistically significant correlation between the individual drug effects on pupil size and on heart rate (S1B Fig), in line with the idea that pupil size reflects central arousal more closely than heart rate, a measure of peripheral arousal.

Atomoxetine did not significantly affect objective task performance. While signal strength lawfully decreased reaction time (RT; $F_{1,18} = 21.7$, $p < 0.001$; two-way repeated measures ANOVA) and increased accuracy ($F_{1,18} = 381.0$, $p < 0.001$) (Fig 2A and 2B), atomoxetine changed neither of those behavioral parameters (RT: $F_{1,18} = 2.1$, $p = 0.160$; accuracy: $F_{1,18} = 2.8$, $p = 0.110$; no significant interaction of signal strength and drug; Figs 2A, 2B and S2A). Likewise, atomoxetine did not significantly change signal detection theoretic sensitivity (d'; S2B Fig).

By contrast, atomoxetine had a robust impact on the impact of history-based (i.e., dynamically varying throughout the course of the experiment) prior expectations on their decisions. We operationalized these time-varying prior expectations as the "repetition probability", the probability of repeating the choice from the previous trial, irrespective of a potential overall bias for the left or right choice options (Materials and methods). The stimulus repetition probability was, by design, close to 0.5 (Fig 2C) (group average: 0.496; Materials and methods), thus not explaining the sequential effects in participants' behavior. Atomoxetine reduced the mean choice repetition probability (Fig 2D; $F_{1,18} = 9.0$, $p = 0.008$; two-way repeated measures ANOVA). Signal strength had no such effect ($F_{1,18} = 0.8$, $p = 0.381$) and signal strength and drug did not interact ($F_{1,27} \sim 0.0$, $p = 0.850$). Qualitatively, we observed the same pattern after correct or incorrect decisions (S2C and S2D Fig), and there was no significant difference between the drug effect on repetition probability after correct and error trials ($t = 0.22$, $p = 0.828$, BF = 0.243).

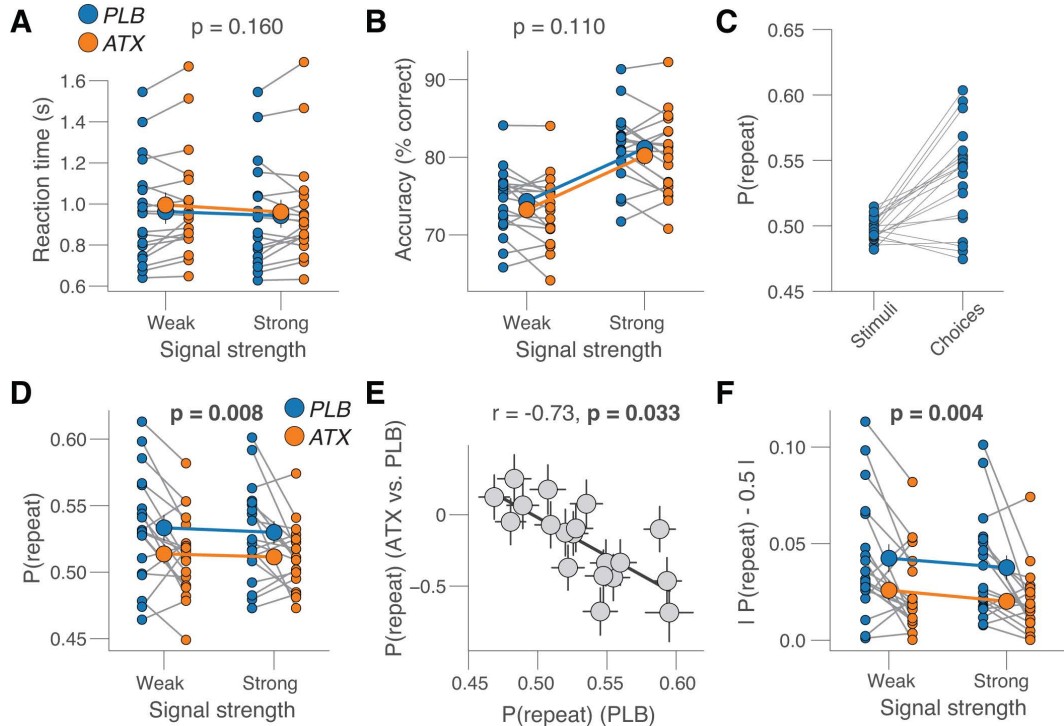

**Fig 2. Atomoxetine reduces choice history bias. (A)** Reaction time, separately for placebo and atomoxetine sessions, and separately or weak and strong signals. Every connecting line is a participant; large data points in the middle are the group averages (error bars, s.e.m. across 19 participants); stats, main effect of drug in two-way repeated measures ANOVA. See S2A Fig for reaction time distributions. **(B)** As A, but for accuracy. **(C)** Probability of repeating the previous stimulus category (left) or choice (right) during placebo sessions. **(D)** As A, but for repetition probability (Materials and methods). **(E)** Individual shift in repetition probability caused by atomoxetine, plotted against individual's repetition probability during the placebo sessions. Data points are individual participants; error bars, 68% confidence intervals across 5K bootstraps; stats, Pearson's correlation coefficient (corrected for reversion to the mean; S2E Fig; Materials and methods). **(F)** As A, but for absolute repetition probability (Materials and methods). https://doi.org/10.5281/zenodo.16779024.

In the placebo sessions, most participants (15 out of 19) had a repetition probability larger than 0.5, that is, they tended to repeat their choices (Fig 2C). We, therefore, wondered if (i) atomoxetine reduced the group-level choice history biases by stereotypically pushing behavior more toward alternation (i.e., reduction of repetition probability, in line with previous pupillometry work [8]) or (ii) through a reduction of both repetition and alternation biases (shift of repetition probability toward 0.5). Our results are in line with the second scenario. There was a robust correlation between individual choice history biases during placebo sessions and the shift in history biases caused by atomoxetine: participants with the strongest repetition biases exhibited the strongest shift towards a repetition probability of 0.5 (Fig 2E). Further, our data are consistent with the idea that atomoxetine also reduced alternation biases, although the latter were only observed in four participants and were weak (Fig 2E, data points on x-axis <0.5). Consequently, atomoxetine yielded an even more consistent and robust reduction of the absolute choice history bias, defined as the absolute value of $P$(repeat)−0.5 (Fig 2F; $F_{1,18} = 11.0$, $p = 0.008$; two-way repeated measures ANOVA). Atomoxetine did not change overall choice bias, quantified as signal detection theoretic criterion (S2F–S2H Fig). These findings show that atomoxetine specifically reduced the impact of history-dependent prior expectation in all participants, regardless of whether they idiosyncratically tended to repeat (i.e., assuming a relatively stable environment) or alternate (assuming a volatile environment).

How did participants' expectations shape their decision processes, and how did atomoxetine act on this mechanism? Current models of perceptual decision-making posit the temporal accumulation of sensory evidence, resulting in

an internal decision variable that grows with time [41,45–47]. When this decision variable reaches one of two decision bounds, a choice is made, and the corresponding motor response is initiated. In this framework (Fig 3A), a bias can arise in two ways: (i) by shifting the starting point of accumulation toward one of the bounds or (ii) by changing the rate of evidence accumulation toward one choice alternative.

Previous work has shown that individual biases in choice repetition probabilities are generally explained by a history-dependent bias in the drift rather than starting point, across a range of different perceptual choice tasks [4]. We replicated this finding for the placebo condition of the present dataset (S3A Fig). To this end, we fitted an accumulation-to-bound model of decision-making to the behavioral data. The model had five free parameters: (1) boundary separation (controlling response caution); (2) the mean drift rate (overall efficiency of evidence accumulation); (3) non-decision time (the speed of predecisional evidence encoding and post-decisional translation of choice into motor response); (4) the starting point of the decision; (5) an evidence-independent bias in the drift (henceforth called "drift bias"). We fitted all five parameters separately per drug condition; additionally, we fitted drift rate separately for both signal strength conditions, and both starting point and drift bias separately for the previous choice category ('left' versus 'right'). The fitted model (S3C Fig) accounted well for the overall behavior in each task (S3D and S3E Fig). Indeed, atomoxetine also specifically reduced the history-dependent shift in drift bias, but not the history shift in starting point (Fig 3B). Furthermore, this drift bias (not starting point) effect was robustly and specifically correlated to the individual change of choice repetition probability (Fig 3C). Across subjects, the history-dependent shift in starting point did not predict a drug effect therein ($r = -0.34$, $p = 0.673$; corrected for reversion to the mean), and the same was true for shifts in drift bias ($r = -0.60$, $p = 0.233$).

Previous studies linked choice history biases to evoked pupil responses during perceptual decisions [8], a likely correlate of the task-evoked responses of locus coeruleus (potentially other brainstem nuclei) during decision-making [24,32,33]. We, therefore, also evaluated the relationship between choice history bias and the pupil measurements obtained in the placebo sessions of the current study. Because our fast-paced task design was not suited to isolate trial-to-trial variations of pretrial baseline pupil size from the task-evoked responses (Materials and methods), we focused this analysis exclusively on the pupil responses. Building on previous work finding the decision-related component of task-evoked pupil responses to be precisely locked to participants' behavioral choice [8,32,48,49], we analyzed the pupil

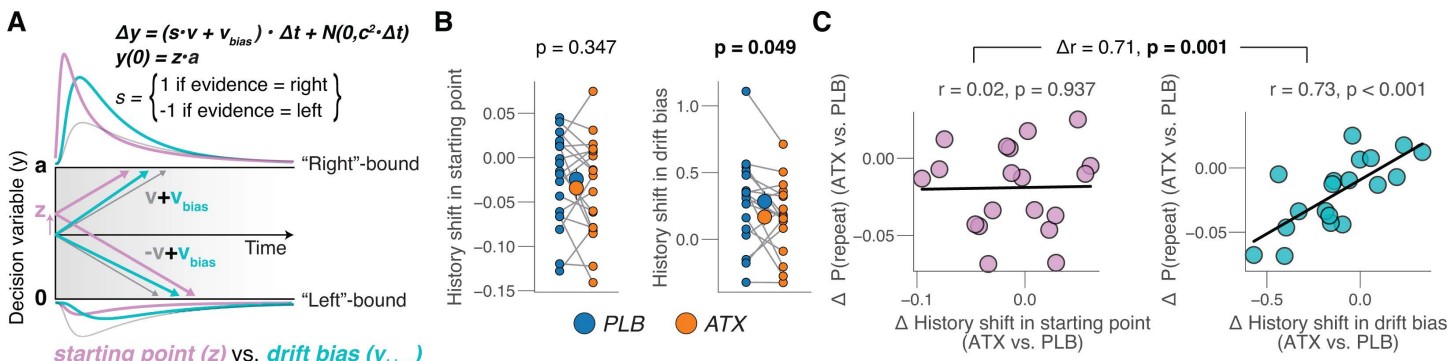

**Fig 3. Atomoxetine reduces choice history bias in evidence accumulation. (A)** Schematic of drift diffusion model accounting for choices, and their associated reaction times (RTs). In the equation, v is the drift rate. Purple and cyan curves show expected RT distributions under shifts in either the 'starting point' (z; purple) or 'drift bias' ($v_{bias}$; cyan). **(B)** Left: history-dependent shift in starting point bias (Materials and methods), separately for placebo and atomoxetine sessions. Right, as left, but for drift bias. Every connecting line is a participant; large data points in the middle are the group averages (error bars, s.e.m. across 19 participants); stats, paired-sample $t$ test. **(C)** Left: individual shift in repetition probability caused by atomoxetine, plotted against individual's history-dependent shift in starting point bias. 'History shift' is the difference between a parameter estimate for previous 'left' and previous 'right' choices, irrespective of the category of the current stimulus. Right: as left, but for history-dependent shift in drift point bias. Data points are individual participants; stats, Pearson's correlation coefficient (difference in correlation assessed with permutation test; Materials and methods). https://doi.org/10.5281/zenodo.16779024.

responses time-locked to the choice report. This showed an early constriction likely related to the appearance of the two salient choice targets of relatively high baseline contrast [50,51]. This response was followed by two successive dilations, whereby the first peaked shortly after the choice report and the second followed about 530 ms later (Fig 4A). We interpret the first peak to reflect the decision-related response [8,32,48] and the second peak to be driven by the auditory feedback [52,53], which in our current task was triggered by the registration of the button press (Materials and methods). Accordingly, we focused our analyses of behavioral correlates of pupil responses on the first peak (gray window in Fig 4A) and completed this by a time-variant analysis (S4A Fig).

We found a negative correlation between the amplitude of the decision-related pupil response component and the probability of repeating the previous choice (Figs 4B and S4A). Likewise, the decision-related pupil response amplitude predicted a reduction of choice repetition bias on the next trial (S4B Fig), replicating previous findings [8]. Both effects mimic the effect of atomoxetine shown in Fig 2, in line with the notion that a boost of central arousal and catecholamine levels at both slow, session-related (pharmacology) and fast, trial-related (pupil responses) timescales have consistent effects on choice history bias. Additionally, task-evoked pupil responses were negatively related to accuracy and quadratically related to RT (S4B Fig).

In a final analysis, we tested if the individual effect of atomoxetine on pupil size, measured as the average across task blocks (Fig 1D), predicted the individual effect of atomoxetine on choice history bias. Participants with a larger increase in pupil size after taking atomoxetine also exhibited larger reductions in repetition probability (Fig 4C). This pattern was not statistically significant for heart rate (S4C Fig). In sum, we here found analogous and correlated effects on choice history bias of pharmacological catecholamine enhancement (via atomoxetine) and of slow and fast variations of pupil size, a marker of central arousal state.

## Discussion

Using selective pharmacological interventions in humans, we here imply catecholamines in the modulation of human choice history biases: boosting catecholamines levels through atomoxetine specifically reduced the magnitude of choice history biases. Catecholamines, and noradrenaline in particular, have previously been implicated in modulating learning rates [54–57] and facilitating set shifting [58] in dynamic environments. Here, we use a selective pharmacological intervention to provide causal evidence for their role in controlling the weight of historical information in perceptual decisions. We

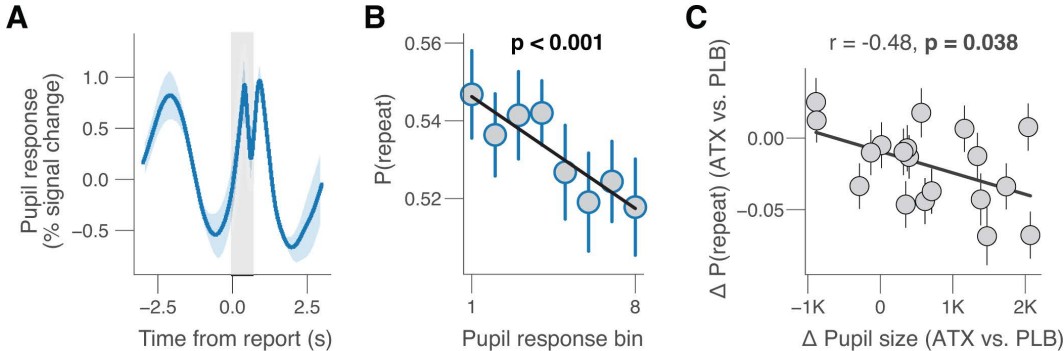

**Fig 4. Pupil-linked arousal reduces choice history bias. (A)** Grand-average pupil response time-locked to report (button press) during placebo. Gray window, interval for averaging task-evoked pupil response on single trials (Materials and methods). **(B)** Relationship between task-evoked pupil response and repetition probability (8 bins) during placebo. Stats, mixed linear modeling (Materials and methods). See S4A Fig for time-wise regression and S4B Fig for other behavioral measures. **(C)** Individual shift repetition probability caused by atomoxetine, plotted against individual shift in pupil size caused by atomoxetine. Pupil size measured as the average across task blocks (Materials and methods), separately for placebo and atomoxetine sessions; data points are individual participants; error bars, 68% confidence intervals across 5K bootstraps; stats, Pearson's correlation coefficient. Panels A and B: shading or error bars, s.e.m. across 19 participants. https://doi.org/10.5281/zenodo.16779024.

interpret our findings in the context of theories postulating a noradrenergic down-weighting of prior expectations (which, we assume, drive the choice history biases analyzed here) in combination with new sensory evidence in the evaluation of current environmental state [22,23,59]. In the current task, participants' history biases were likely due to an implicit belief that the task environment was relatively stable, i.e., that stimuli presented on successive trials are correlated. Our finding that atomoxetine lowered this bias is thus in line with the above theoretical frameworks [22,23,59]. Our findings are also consistent with the idea that participants attended to one side of the screen for several successive trials, producing choice "streaks". Thus, the drug effects may also reflect reduced hysteresis of attentional selection. While this scenario is not mutually exclusive with the above theoretical accounts, future research should develop approaches specifically tailored to isolating the dynamics of attentional selection (and associated drug effects) in choice history biases.

Our finding of a causal effect of catecholamine levels on history bias is consistent with correlative evidence from pupillometry, obtained in previous work [8,60–64] and replicated here. In dynamic environments, pupil responses are associated with violations of learned, top-down expectations and reduce their weight [60–64]. In stable environments, pupil responses predict a reduction of choice repetition probability [8]. There are, however, also some subtle but notable differences between our current atomoxetine effects and pupil-linked effects on behavior. First, one study found the pupil-predicted reduction of repetition probability to be a general increase in the tendency to alternate [8], while we found atomoxetine to reduce history biases regardless of their sign (i.e., promoting a shift of repetition probability toward 0.5), which is more consistent with a reduction in prior weight. Second, our pharmacological intervention did not affect participant's overall choice bias (bias towards one choice alternative, irrespective of the immediate history), even though this is another commonly reported behavioral correlate of pupil responses [32,48,65–67] Third, in our current dataset, pupil responses exhibited a monotonic (negative) relationship to accuracy, without any concomitant accuracy effect of atomoxetine. One possible explanation for these differences is that atomoxetine likely increased both tonic and phasic catecholamines levels [68], while task-evoked pupil responses reflect phasic neuromodulatory activity [32]. Another possibility is that while atomoxetine selectively elevates central catecholamine levels (noradrenaline and dopamine), pupil responses track not only noradrenaline [31,32,42,43,69,70], but also acetylcholine [31,71], serotonin [72], and orexin [73] levels. Even so, our current analyses of pharmacological and pupil-related effects of the same data set indicate overall analogous effects of the variations in catecholaminergic tone at different timescales on choice history biases.

Which neuromodulator specifically mediated the effects reported here? Atomoxetine increases the extracellular levels of noradrenaline, but also of (prefrontal) dopamine (Materials and methods). Indeed, one recent report suggests that striatal dopamine may play a role in adapting choice history bias to temporal regularities in a stimulus sequence [13]. Thus, our results are consistent with a role of one or both catecholamines. However, the noradrenergic locus coeruleus and dopaminergic midbrain exhibit a close interplay [74,75] and are co-activated with pupil responses during decisions [32]. Thus, it could well be that their (complex) interaction regulates choice history bias in perceptual decisions.

In sum, pharmacologically elevating central catecholamine (specifically noradrenaline) levels in human participants specifically reduces their propensity to accumulate noisy evidence for perceptual decisions in a manner that is biased by previously reported perceptual experience. This supports computational theories according to which noradrenaline reduces the impact of dynamic priors on inference and decision-making.

## Materials and methods

### Participants

Nineteen participants (14 female, age range 20–26) participated in the experiment, entailing concurrent pupillometry and magnetoencephalography (MEG) recordings. All had normal or corrected to normal vision and no history or indications of psychological or neurological disorders. All participants gave written informed consent. The experiment was approved by the ethics committee of the University Medical Center Hamburg-Eppendorf (approval number, PV4843) and conducted in accordance with the Declaration of Helsinki. Participants participated in four experimental sessions, in which we crossed

drug condition (atomoxetine versus placebo) and stimulus-response mapping (press ipsi versus contra) (about 2 h per session). Participants were paid 10€ per hour. One participant completed three recording sessions.

**Pharmacological intervention**

We used the selective noradrenaline reuptake inhibitor atomoxetine (dose, 40 mg) to boost the levels of catecholamines, specifically noradrenaline and (in the prefrontal cortex) dopamine [34,35]. Atomoxetine is a relatively selective inhibitor of the noradrenaline transporter, which is responsible for the natural reuptake of noradrenaline that has been released into the extracellular space. Consequently, atomoxetine acts to increase the extracellular levels of noradrenaline, an effect that has been confirmed experimentally in rat prefrontal cortex [34]. The same study showed that atomoxetine also increases the prefrontal levels of dopamine, which has a molecular structure very similar to the one of noradrenaline and is, in fact, a direct precursor of noradrenaline. Atomoxetine has smaller affinity to the serotonin transporter, and there are discrepant reports about the quantitative relevance of these effects: while one study found no increases in serotonin levels under atomoxetine [34], a recent study reports a significant atomoxetine-related occupancy of the serotonin transporter in non-human primates [76] at dosages that would correspond to human dosages of 1.0–1.8 mg/kg. Note that these dosages are substantially higher than the administered dosage in this study (40 mg, independent of body weight). It is therefore unclear to what extent our atomoxetine condition affected cortical serotonin levels. A mannitol-aerosil mixture was administered as placebo. All substances were encapsulated identically to render them visually indistinguishable. Peak plasma concentrations are reached about 1–2 h after administration for atomoxetine, and the half-life is about 5 h [77]. Thus, participants received either a placebo or atomoxetine 1.5 hours before the start of the experimental session.

**Behavioral task**

Each trial consisted of three consecutive intervals (Fig 1B): (i) the baseline interval (uniformly distributed between 0.6 and 1.4 s) containing a dynamic noise pattern on both the left and right side of the green fixation dot; (ii) the decision interval (terminated by the participant's response; max 3 s) containing the same noise patterns with Gabor patch embedded in each; (iii) the feedback interval (uniformly distributed between 0.25 and 1.4 s). Each block consisted of 160 trials, with mini-breaks after every 40 trials.

Two dynamic noise patterns were presented throughout the experiment. The luminance across all pixels was kept constant. This pedestal noise pattern had 20% contrast. In the decision interval, two Gabor patches (sinusoidal gratings; 2 cycles per degree; phase randomly drifted left or right) were added to each noise pattern. The Gabor patches symmetrically differed from 80% contrast. On half of trials, the Gabor patch on right side of the screen was of higher contrast (signal, right); on the other half the Gabor patch on the left side was of higher contrast (signal, left). Signal location was randomly selected on each trial, under the constraint that it would occur on 50% of the trials within each block of 160 trials. Stimuli were presented within two circular patches to the left and right of the central fixation cross (diameter, 15° of visual angle; eccentricity, 12°).

Participants were instructed to report the location of the signal, the highest contrast Gabor patch by pressing one of two response buttons with their left or right index finger as soon as they felt sufficiently confident ("free response paradigm"). We did not give specific instructions to tradeoff accuracy for speed (or vice versa). The mapping between perceptual choice and button press (e.g., "right" —> press right key; "left" —> press left key) was counterbalanced across the four recording sessions. Participants received auditory feedback (high tone, correct; low tone, error) 50 ms after they indicated their choice.

Signal strength, the difference in contrast between the left and right signals, was individually titrated before the main experiment to two difficulty levels that yielded about 70% and 85% correct choices, using an adaptive staircase procedure (Quest) [78]. In the placebo sessions, we measured a mean accuracy of 74.07% correct (± 0.97% s.e.m.) and 80.83% correct (± 1.12% s.e.m), for weak and strong signal strengths, respectively.

Stimuli were back-projected on a transparent screen using a Sanyo PCL-XP51 projector with a resolution of 800 × 600 at 24 Hz. The screen was positioned 65 cm away from their eyes. The luminance profile was linearized by measuring and correcting for the systems gamma curve. A doubling of contrast values, therefore, also produced a doubling of luminance differences. During the first training session stimuli were presented on a VIEWPixx monitor with the same resolution and refresh rate (also linearized). To minimize any effect of light on pupil diameter, the overall luminance of the screen was held constant throughout the experiment.

Participants performed between 22 and 27 blocks (distributed over four sessions) yielding a total of 3,520–4,320 trials per participant. One participant performed a total of 17 blocks (distributed over three scanning sessions), yielding a total of 2,720 trials.

## Data acquisition

Recordings took place in a dimly lit magnetically shielded room. MEG was recorded using a whole-head CTF 275 MEG system (CTF Systems, Canada) at a sampling rate of 1,200 Hz. In addition, eye movements and pupil diameter were recorded with an MEG-compatible EyeLink 1000 Long Range Mount system (SR Research, Osgoode, ON, Canada). Finally, electrocardiogram (ECG) as well as vertical and horizontal electrooculogram were acquired using Ag/AgCl electrodes at a sampling rate of 1,200 Hz. MEG data will be the focus of a different report.

## Analysis of electrocardiogram data

ECG data were used to analyze average heart rate. We used an adaptive threshold to detect the R-peak of each QRS-complex in the ECG. Heart rate was then computed by dividing the total number of R-components by time. We quantified block-wise heart rate as the average heart rate across entire task blocks.

## Analysis of pupil data

All analyses were performed using custom-made Python scripts, unless stated otherwise.

**Preprocessing.** Periods of blinks and saccades were detected using the manufacturer's standard algorithms with default settings. The remaining data analyses were performed using custom-made Python scripts. We applied to each pupil timeseries (i) linear interpolation of missing data due to blinks or other reasons (interpolation time window, from 150 ms before until 150 ms after missing data), (ii) low-pass filtering (third-order Butterworth, cut-off: 6 Hz), (iii) removal of pupil responses to blinks and to saccades, by first estimating these responses by means of deconvolution and then removing them from the pupil time series by means of multiple linear regression [79], and (iv) conversion to units of modulation (percent signal change) around the mean of the pupil time series from each measurement session.

**Quantification of block-wise pupil size.** We quantified block-wise pupil size as the average blink-interpolated pupil size (in raw Eyelink units) across entire task blocks.

**Quantification of task-evoked pupil responses.** We quantified task-evoked pupil responses for each trial as the mean of the pupil size (in units of percent signal change) in the window 0–0.65 s from choice (gray window in Fig 4A), minus the pretrial pupil size, measured as the mean pupil size during the 0.25 s before trial onset.

## Analysis and modeling of choice behavior

All analyses were performed using custom-made Python scripts, unless stated otherwise. Trials in which participants failed to respond within 3 s were excluded from the analyses. Additionally, we excluded trials for which no previous choice was available: (i) trials after which participants failed to respond within 3 s, (ii) first trial of each block, and (iii) first trial after each mini-break. We always computed behavioral metrics separately for each participant and drug condition.

**Overt behavior.** RT was defined as the time from stimulus onset until the button press. Repetition probability was defined as the fraction of choices that was the same as the choice on the previous trial, irrespective of a potential overall side-bias:

$$P(repeat) \ = \ \frac{P(R_n = left | R_{n-1} = left) + P(R_n = right | R_{n-1} = right)}{2} \tag{1}$$

where $R$ is the response (left or right) and $n$ is trial number.

**Signal-detection theoretic modeling.** The signal detection [80] metrics sensitivity ($d'$) and criterion ($c$) were computed separately for each of the pupil bins. We estimated $d'$ as the difference between $z$-scores of hit-rates and false-alarm rates. We estimated the criterion by averaging the $z$-scores of hit-rates and false-alarm rates and multiplying the result by −1.

**Drift diffusion modeling.** We fitted the reaction time data (separately per session) with the drift diffusion model [45,47], based on continuous maximum likelihood using the Python-package PyDDM [81].

The decision dynamics were described by (Fig 3A):

$$\Delta y \ = \ (\ s \cdot v + v_{bias}) \cdot \Delta t \ + \ cdW \tag{2}$$

where $y$ is the decision variable, $s$ is the stimulus category (−1 for left signals; 1 for right signals), $v$ is the drift rate and controls the overall efficiency of evidence accumulation, $v_{bias}$ is the drift bias which is an evidence-independent constant added to the drift, and $cdW$ is Gaussian distributed white noise with mean 0 and variance $c^2 \Delta t$. The starting point of evidence accumulation $z$ was defined as a fraction of the boundary separation:

$$y(0) \ = \ z \cdot a \tag{3}$$

where $a$ is the separation between the two decision bounds. Evidence accumulation terminated at 0 ("left") or $a$ ("right"). The model was fitted separately for each recording session of each participant. In each fit, we let drift rate vary with signal strength [47], and starting point and drift bias with previous choice [4].

## Statistical comparisons

We used a paired sample $t$ test to test for significant differences in heart rate, pupil size, and behavioral measures between placebo and drug sessions (Figs 1 and 3). We used 2 × 2 repeated measures ANOVA to test for the main effects of drug and signal strength, as well as their interaction, on behavioral measures (Fig 2). All tests were performed two-tailed.

We performed permutation analyses to obtain the distribution of correlation coefficients predicted exclusively by regression to the mean (Fig S2E). This was done by computing the above-described across-subject correlation 10K, each time using randomly assigned "placebo" versus "drug" labels of each participant. We then compared our observed correlation coefficient (reflecting the combined effects of regression to the mean and the relationship between overall bias and the atomoxetine predicted shift therein) against this permutation distribution. This analysis suggested that the relationship between choice history bias and the atomoxetine predicted shift therein was stronger than expected based on regression to the mean (proportion of permutations below observed correlation <0.05).

We also used a permutation procedure to test for differences between the correlation coefficients for history shifts in starting point bias and drift bias (Fig 3C). Again, we permuted the condition labels (drug/placebo).

We used a mixed linear modeling approach implemented in the Python-package 'Statsmodels' [82] to quantify the dependence of several metrics of overt behavior on pupil response bin. Our approach was analogous to sequential

polynomial regression analysis [83], but now performed within a mixed linear modeling framework. We fitted two mixed models to test whether pupil response bin predominantly exhibited a monotonic effect (first-order) or a non-monotonic effect (second-order) on the behavioral metric of interest (y). The fixed effects were specified as:

$$\text{Model 1}: \ y \sim \beta_0 1 + \ \beta_1 P \tag{4}$$

$$\text{Model 2}: \ y \sim \beta_0 1 + \ \beta_1 P + \ \beta_2 P^2 \tag{5}$$

with $\beta$ as regression coefficients and $P$ as the task-evoked pupil response bin number (demeaned). We included random intercepts (intercepts could vary with participant). The mixed models were fitted through restricted maximum likelihood estimation. Each model was then sequentially tested in a serial hierarchical analysis, which was based on BIC. This analysis was performed for the complete sample at once, and it tested whether adding the next higher order model yielded a significantly better description of the response than the respective lower order model (difference in BIC >10). We tested models from the first-order (constant, no effect of pupil response) up to the second-order (quadratic, non-monotonic).

## Supporting information

**S1 Fig.** **(A)** Schematic of stimulus-response mapping rules. **(B)** Individual shift in pupil size caused by atomoxetine, plotted against individual shift in heart rate caused by atomoxetine. Heart rate and pupil size measured as the average across task blocks (Materials and methods), separately for placebo and atomoxetine sessions (Materials and methods); data points are individual participants; stats, Pearson's correlation coefficient. https://doi.org/10.5281/zenodo.16779024.
(EPS)

**S2 Fig. Atomoxetine does not affect overall choice bias. (A)** Histograms of reaction times, separately for placebo and atomoxetine sessions. **(B)** Signal detection theoretic sensitivity (Materials and methods), separately for placebo and atomoxetine sessions, and separately or weak and strong signals. Every connecting line is a participant; large data points in the middle are the group averages (error bars, s.e.m. across 19 participants); stats, main effect of drug in two-way repeated measures ANOVA. **(C)** As B, but for repetition probability after correct choices (positive feedback). **(D)** As C, but after incorrect choices (negative feedback). **(E)** Permutation (randomized) distribution of correlation coefficients describing relationship between individual shift in repetition probability caused by atomoxetine and individual's repetition probability during the placebo sessions. The observed correlation (plotted in Fig 2E) is stronger than 99% of this permutation distribution, resulting in a $p$-value of 0.01. **(F)** As B, but for signal detection theoretic criterion (Materials and methods). **(G)** Individual shift in criterion caused by atomoxetine, plotted against individual's criterion during the placebo sessions. Data points are individual participants; error bars, 68% confidence intervals across 5K bootstraps; stats, Pearson's correlation coefficient (corrected for reversion to the mean; Materials and methods). **(H)** As F, but for absolute criterion (Materials and methods). https://doi.org/10.5281/zenodo.16779024.
(EPS)

**S3 Fig. Individual choice history biases are better explained by history-dependent changes in drift bias than starting point bias. (A)** Relationship between individual choice repetition probabilities [$P$(repeat)] and history shift in starting point (left column) and drift bias (right column), in the placebo sessions. Parameter estimates were obtained from a model in which both bias terms were allowed to vary with previous choice (Materials and methods). Stats, Pearson's correlation coefficient. The plots show that the starting point was shifted away from the previous response for most participants (negative values along x-axis in left panel) and drift bias was shifted towards the previous response (positive values along x-axis in right panel. These opposing effects have been reported before [4]. **(B)** As A, but for atomoxetine sessions. **(C)** Group-level boxplots of all parameter estimates. Black diamonds, outliers. **(D)** Measured and predicted RT

distributions, separately for 'left' and 'right' choices, across all participants (left column) and conditional response functions (right column)), in the placebo sessions. For the conditional response functions, we divided all measured and model-predicted trials into five quantiles of the RT distribution and plotted for each quantile the fraction of 'right'-choices. Error bars, s.e.m. across 19 participants. **(E)** as D, but for atomoxetine sessions. https://doi.org/10.5281/zenodo.16779024. (EPS)

**S4 Fig.** **(A)** Beta values from time-wise regression of repetition probability and task-evoked pupil response (same relationship as in Fig 4B). **(B)** Relationship between task-evoked pupil response (same time-window as used main Fig 4; Materials and methods) and accuracy (left), RT (middle) or repetition probability w.r.t next trial (right). Stats, mixed linear modeling (Materials and methods). **(C)** Individual shift repetition probability caused by atomoxetine, plotted against individual shift in heart rate caused by atomoxetine. Heart rate measured as the average across task blocks (Materials and methods), separately for placebo and atomoxetine sessions; data points are individual participants; error bars, 68% confidence intervals across 5K bootstraps; stats, Pearson's correlation coefficient. https://doi.org/10.5281/zenodo.16779024. (EPS)

## Author contributions

**Conceptualization:** Jan Willem de Gee, Niels A. Kloosterman, Tobias H. Donner.

**Data curation:** Jan Willem de Gee, Niels A. Kloosterman.

**Formal analysis:** Jan Willem de Gee, Niels A. Kloosterman, Anke Braun.

**Funding acquisition:** Tobias H. Donner.

**Investigation:** Jan Willem de Gee, Niels A. Kloosterman.

**Project administration:** Tobias H. Donner.

**Supervision:** Tobias H. Donner.

**Writing – original draft:** Jan Willem de Gee, Tobias H. Donner.

**Writing – review & editing:** Jan Willem de Gee, Niels A. Kloosterman, Anke Braun, Tobias H. Donner.

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
