## [Editor Report · Decision Letter 0]

7 Nov 2024

Dear Dr de Gee,

Thank you for submitting your manuscript entitled "Catecholamines reduce choice history biases" for consideration as a Research Article by PLOS Biology.

Your manuscript has now been evaluated by the PLOS Biology editorial staff as well as by an academic editor with relevant expertise and I am writing to let you know that we would like to send your submission out for external peer review.

Once your full submission is complete, your paper will undergo a series of checks in preparation for peer review. After your manuscript has passed the checks it will be sent out for review. To provide the metadata for your submission, please Login to Editorial Manager (https://www.editorialmanager.com/pbiology) within two working days, i.e. by Nov 09 2024 11:59PM.

Kind regards,

Christian

Christian Schnell, PhD

Senior Editor

PLOS Biology

cschnell@plos.org

---

## [Decision Letter · Decision Letter 1]

16 Jan 2025

Dear Dr de Gee,

Thank you for your patience while your manuscript "Catecholamines reduce choice history biases" was peer-reviewed at PLOS Biology. It has now been evaluated by the PLOS Biology editors, an Academic Editor with relevant expertise, and by several independent reviewers.

In light of the reviews, which you will find at the end of this email, we would like to invite you to revise the work to thoroughly address the reviewers' reports.

As you will see below, the reviewers are interested in your study and are overall supportive, but raise a number of concerns that need to be addressed. Reviewer 1 raises a number of major concerns regarding the specificity of the effects (peripheral effects of atomoxetine), discrepancies with earlier studies, bias through attentional processes in stimulus selection, and internal contradictions of hypotheses and results. Reviewer 2's concerns are overall less critical, but will require additional analyses and textual revisions to address.

Given the extent of revision needed, we cannot make a decision about publication until we have seen the revised manuscript and your response to the reviewers' comments. Your revised manuscript is likely to be sent for further evaluation by all or a subset of the reviewers.

**IMPORTANT - SUBMITTING YOUR REVISION**

*Re-submission Checklist*

*Published Peer Review*

*PLOS Data Policy*

*Blot and Gel Data Policy*

Sincerely,

Christian

Christian Schnell, PhD

Senior Editor

PLOS Biology

cschnell@plos.org

REVIEWS:

Reviewer #1 (Rei Akaishi)

The authors have conducted an exemplary study examining catecholamine effects on choice behavior, specifically focusing on tendencies to either repeat or alternate choices. This well-written paper addresses a topic of central importance to cognitive neuroscience. The study involved administering atomoxetine to subjects before they performed standard perceptual decision-making tasks with two visual stimuli. The study's conception is well-grounded in theoretical ideas about catecholamines and previously observed effects of arousal on choice repetition tendencies. While the authors' ideas appear clear initially, there are several alternative interpretations of the results and some inconsistencies between current and previous theoretical frameworks. Additionally, there are some conceptual ambiguities and interpretational issues within the current study. The authors should address these concerns to strengthen the paper.

Major Issues:

1. While the authors attribute atomoxetine's effects on choice repetition/alternation to its central action on the brain's norepinephrine system, the orally administered drug also affects peripheral systems. Indeed, the authors report acute effects on heart rate. This raises the possibility that the observed behavioral effects stem from peripheral changes that subsequently influence central information processing. The authors should examine relationships between peripheral effects (heart rate and pupil size) and choice tendencies, although control analyses were performed for RT and accuracy. Additional measures of cognitive function would help establish the drug's central effects, particularly given that atomoxetine's acute effects are predominantly peripheral.

2. Assuming central effects are present, there is a theoretical discrepancy between this study and Dayan and Yu's previous work on catecholamine effects in the central nervous system. The Dayan-Yu framework focuses not on norepinephrine's repetition effects per se, but on cue relevance among multiple cues. This conceptual difference requires explanation.

3. The task design is relevant here, as cue-based attention could select one of the Gabor patches and sustain choice repetition. Thus, the observed drug effects may reflect attentional processes in stimulus selection rather than purely decision processes.

4. There are concerns about the study's hypotheses and testing methods. Hypothesis (ii) appears ad-hoc. The authors should clarify whether these hypotheses were preregistered.

5. The support for hypothesis (ii) in Figure 2E is somewhat weak, as the strengthening of alternation by atomoxetine is not clearly demonstrated. While data show that stronger repetition biases are reduced toward 0.5 choice probability with atomoxetine, the statement about correlation between placebo session biases and atomoxetine-induced shifts requires stronger evidence.

6. The use of the drift diffusion model to examine atomoxetine's effect on choice repetition appears inconsistent with the authors' earlier support for hypothesis (ii). Hypothesis (ii) addresses both repetition and alternation, yet these separate analyses yield seemingly contradictory results that need reconciliation.

Reviewer #2: This study investigates the effect of catecholamines on perceptual choice repetition biases. The authors found that administering the noradrenaline reuptake inhibitor atomoxetine reduced choice history biases compared to placebo. Further, by fitting drift diffusion models the authors found that this was due to a reduction in history-dependent shifts of drift rate rather than the starting point of evidence accumulation. Together, the results are taken as evidence that catecholamines reduce the impact of a specific form of priors (choice history) on perceptual decisions.

The study addresses a timely topic of perceptual choice history biases and contributes to our understanding of the role of neuromodulatory systems in perceptual decision-making. The methodology appears sound, the data carefully analyzed, and the manuscript well written. I have a few comments that will hopefully help to improve the manuscript, but otherwise I think it will be a great addition to the literature.

Major comment:

1. The authors quite strongly emphasize the point that atomoxetine reduces choice history biases *regardless of whether those promoted choice repetition or alternation*. However, 15 out of 19 participants exhibit choice repetition, and the remaining 4 participants' repetition probability is fairly close to 50%. In other words, there are no prominent alternation biases in the current data. I understand that the correlation in Figure 2E would imply that if there were participants with pronounced alternation biases, as observed previously, they would be expected to show a reduction towards 50%. I think this extrapolation would be a fair assumption to make, but this assumption should be made more explicit in the Results and Methods.

Minor comments:

2. The authors show that atomoxetine reduces the history shift in drift bias (Fig. 3B). From the analyses it is not entirely clear whether this is a reduction towards zero shift, or a change towards opposite (alternating) shifts. The strong correlation between the effect of drug on P(repeat) and drift bias (Fig. 3C) together with the analyses of P(repeat) in Fig. 2E would suggest that the former is the case, but the authors could repeat their correlation analysis (Fig. 2E) for drift to directly show that participants with larger drift biases in the Placebo condition show the strongest reduction towards zero drift bias in the drug condition.

3. Have the authors looked at the effect of drug on choice repetition after correct and error trials? This would be particularly relevant when discussing differences between the current results and previous results involving pupil size. In a previous paper, it was found that the interaction between pupil size and choice repetition was opposite following correct and error trials, and resulted in a net repulsion bias (Supplementary Fig. 11, Urai et al., 2017). Is a similar gating by previous outcome present in the current data, perhaps with a dominance of post-error correction that reduces choice repetition towards neutral?

4. Related to the previous point, pupil size was recorded in the current study (Fig. 1D). This seems like a fantastic opportunity to compare effects of pupil size during the placebo sessions with that of atomoxetine during drug sessions. If the authors find evidence that pupil size promotes choice alternation, this within-participant difference to the effect of atomoxetine would provide an even more convincing basis for the discussion of the effects of tonic vs. phasic catecholamine levels.

5. I was wondering how P(repeat) was calculated? The authors write: "Repetition probability was defined as the fraction of choices that was the same as the choice

on the previous trial." Does this mean P(repeat) = P(Response(t) = Response(t-1))? Or is P(repeat) the average between p(Response(t) = Left | Response(t-1) = Left) and p(Response(t) = Right | Response(t-1) = Right)? If I'm not mistaken, the former definition would include history-independent side biases in P(repeat). I would appreciate if this would be described in more detail.

6. For the computational interpretation, I am wondering whether one can distinguish down-weighting of prior expectations from weaker updating of priors. Especially since catecholamines have been implicated in modulating learning rates, it seems possible that reduced history biases stem from a weaker updating of priors rather than down-regulating the impact of priors. Is this a distinction the authors would like to make? I think it would be worthwhile to discuss this in more detail.

7. P.8, line 348: "Reaction time (RT) was defined as the time from stimulus offset until the button press." Should this be "stimulus onset"?

8. P.9, line 387: The authors write "pupil predicted shift". I think this should say "Atomoxetine predicted shift".

---

## [Decision Letter · Decision Letter 2]

23 Jul 2025

Dear Jan Willem,

Thank you for your patience while we considered your revised manuscript "Catecholamines reduce choice history biases" for publication as a Research Article at PLOS Biology. This revised version of your manuscript has been evaluated by the PLOS Biology editors, the Academic Editor and two of the original reviewers.

Based on the reviews and on our Academic Editor's assessment of your revision, we are likely to accept this manuscript for publication, provided you satisfactorily address the remaining points raised by the reviewers. Please also make sure to address the following data and other policy-related requests:

* We would like to suggest a different title to improve its accessibility for our broad audience: "Catecholamines reduce choice history biases in perceptual decision making"

* Please add the links to the funding agencies in the Financial Disclosure statement in the manuscript details.

* Please include information in the Methods section whether the study has been conducted according to the principles expressed in the Declaration of Helsinki.

* Please note that per journal policy, the model system/species studied should be clearly stated in the abstract of your manuscript

* DATA POLICY:

Regardless of the method selected, please ensure that you provide the individual numerical values that underlie the summary data displayed in the following figure panels as they are essential for readers to assess your analysis and to reproduce it: 1CD, 2ABCDF, 3B, S2BCDFH and S3C

* CODE POLICY

We expect to receive your revised manuscript within two weeks.

*Published Peer Review History*

*Press*

Sincerely,

Christian

Christian Schnell, PhD

Senior Editor

cschnell@plos.org

PLOS Biology

Reviewer remarks:

Reviewer #1 (Rei Akaishi, signed his report): The authors have adequately addressed the concerns raised in my previous review. However, I note that the mechanisms underlying catecholamine effects on choice repetition may be more complex than what can be fully captured by the current experimental design. In particular, future investigations should carefully examine the implications of the Yu-Dayan theoretical framework for understanding choice repetition bias. Such studies would also contribute to a more nuanced understanding of the underlying mechanisms driving choice repetition phenomena.

Reviewer #2: The authors have carefully addressed my points and I have no further issues.

Minor comment:

I think on page 4, line 143, the authors mean to refer to Figure 2E, not S2E.

---

## [Editor Report · Decision Letter 3]

12 Aug 2025

Dear Jan Willem,

Thank you for the submission of your revised Research Article "Catecholamines reduce choice history biases in perceptual decision making" for publication in PLOS Biology. On behalf of my colleagues and the Academic Editor, Matthew Rushworth, I am pleased to say that we can in principle accept your manuscript for publication, provided you address any remaining formatting and reporting issues. These will be detailed in an email you should receive within 2-3 business days from our colleagues in the journal operations team; no action is required from you until then. Please note that we will not be able to formally accept your manuscript and schedule it for publication until you have completed any requested changes.

PRESS

We frequently collaborate with press offices. If your institution or institutions have a press office, please notify them about your upcoming paper at this point, to enable them to help maximize its impact. If the press office is planning to promote your findings, we would be grateful if they could coordinate with biologypress@plos.org. If you have previously opted in to the early version process, we ask that you notify us immediately of any press plans so that we may opt out on your behalf.

Sincerely, 

Christian

Christian Schnell, PhD

Senior Editor

PLOS Biology

cschnell@plos.org